# Influence of Temperature and Moisture Content on Pavement Bearing Capacity with Improved Subgrade

**DOI:** 10.3390/ma12233826

**Published:** 2019-11-21

**Authors:** Audrius Vaitkus, Laura Žalimienė, Jurgita Židanavičiūtė, Daiva Žilionienė

**Affiliations:** 1Road Research Institute, Vilnius Gediminas Technical University, Linkmenų g. 28, 08217 Vilnius, Lithuania; audrius.vaitkus@vgtu.lt (A.V.); jurgita.zidanaviciute@vgtu.lt (J.Ž.); 2Department of Roads, Vilnius Gediminas Technical University, Saulėtekio al. 11, 10223 Vilnius, Lithuania; daiva.zilioniene@vgtu.lt

**Keywords:** asphalt concrete layer, bearing capacity, deflection, moisture content, road pavement structure, temperature

## Abstract

Environmental conditions (temperature, moisture and the intensity of the sun) influence variation in asphalt pavement strength during the year. Lithuania is situated in a zone by average warm summers and average cold winters, and the most important climatic factor is the variation of the air temperature. This study presents the influence of temperature (of asphalt concrete (AC) and subgrade layers) and moisture content (of subgrade layers) to the pavement bearing capacity. The experimental research was obtained in five pavement sections of the experimental road. This experimental road was constructed in 2007 in Lithuania and is operated for more than 12 years. This paper presents a statistical analysis between the bearing capacity and the thickness of the asphalt concrete layers, the temperature and moisture content of different pavement layers, among sections, loaded and unloaded lanes (right and left wheel paths and tracks). The bearing capacity was evaluated by a falling weight deflectometer (FWD), temperature and moisture content by electronic sensors and thickness of AC layers by Georadar. Analysis of overall *E*_0_ and *E*_0,h9_ (bearing capacity at a depth of 9 cm from asphalt surface) data declares that seasonal impact on pavement structural strength due to a change of subgrade bearing capacity remains after correction of asphalt stiffness dependent on the temperature in the layer. However, it was detected that neither *E*_0_ nor *E*_0,h9_ are related to moisture content at a depths of 100 cm, 130 cm and 150 cm.

## 1. Introduction

Designing pavements according to the present design method assuring and fulfilling construction conditions provides a strength pavement structure (PS) for a given traffic load during a design period under a given climate condition [1].

In addition, Fonod [1] mainly characterizes the serviceability of the pavement:The roughness of the pavement affects driving comfort;Skid resistance affects safe driving;Bearing capacity of the road pavement affects road pavement service life;Structural condition (cracking and patching) affects driving comfort in addition to aesthetic effects.

The bearing capacity of pavement is defined as a characteristic of its structural state, related to the capacity to support the expected traffic. The evaluation of this capacity can be made through parameters such as residual life, required thickness for reinforcement, stiffness modulus and deflection [2].

According to Straube and Jansen [3], the bearing capacity of AC pavements is evaluated by a FWD (the equivalent elastic modulus—*E*_0_). Therefore, the elastic modulus of the AC layers is influenced by the load frequency and AC layer temperature. Whereas, the elastic modulus of the subgrade is influenced by the current water content of the unbound layers and the temperature of the AC layer in case of air temperature below 0 °C.

By Salour [4], solar radiation of the AC surfacing and geothermal heat flux contribute to pavement heat intake, and surface convection and radiation emissions contribute to heat extraction [5]. 

Alawi and Helal [6] results show that the maximum pavement temperature is not on the surface, but rather at a full depth from the surface. This type of temperature change explained due to the heat absorption by the asphalt concrete inside the pavement, and due to air temperature and wind action on the pavement surface. Pavement temperatures remain in the range of 35–50 °C for most of the year. They noticed that temperatures (>68 °C) are attained at a depth of 20 mm and then the temperatures decrease in each layer to its normal distribution. Therefore, the thickness of the asphaltic pavement must be taken into consideration, increasing indirect tensile stiffness modulus for the upper layers of the asphaltic pavement and using materials less affected by high temperature.

According to Motiejūnas et al. [7] based on the analysis of experimental investigation results it is recommended to measure pavement deflections by the falling weight deflectometer when the temperature of asphalt pavement layers varies within the interval +5 °C to +25 °C, except cases when special investigations are carried out during winter freeze and spring thaw periods. Under higher temperature +30 °C to +35 °C the layers of asphalt pavement lose part of their smoothness.

García and Castro [8] studied the effect of temperature on the pavement deflection based on seven deflection tests, at different temperatures, in a section of road with the flexible pavement. The statistical analysis showed a good agreement between the experimental and theoretical obtained factors. However, the influence of temperatures on pavement deflections through experimental factors (with flexible pavements composed of 250 mm of asphalt mixes in a test that representative asphalt temperature range between 15° and 25 °C) is slightly greater in comparison with theoretical factors.

Zheng et al.’s [9] study showed that there is a particular relationship between the pavement thickness and the temperature correction coefficient of the asphalt pavement deflection. With the increase in asphalt pavement thickness, the correction coefficient change rate speeds up. The correction coefficient of 70 mm asphalt pavement thickness is 1.11 at a low temperature of 0 °C and 0.71 at high temperature of 50 °C, while the values are 1.29 and 0.40, respectively for 250 mm asphalt pavement thickness, which indicates that high attention should be given to the temperature correction of asphalt pavement thickness deflection. With the increase in asphalt pavement thickness, the changing trend among temperature correction coefficients becomes stable, for example, the difference of temperature correction coefficients between 150 and 250 mm asphalt pavement thickness deflection is minimal.

As surfacing temperature depends significantly on the weather, typically changing hourly and daily, the physical process never reaches a steady-state, i.e., there would be only a long-term equilibrium [10].

It is known that high temperatures of pavement allow the development of rutting and structural damage during hot and cold periods [11,12].

In seasonal frost conditions, load restrictions are commonly imposed during the spring-thaw period to prevent severe pavement deterioration. During this period, the PSs are usually exposed to excess moisture content, which results in reduced bearing capacity of unbound layers and high resilient and permanent deformations [4].

Precipitation (rain) is the primary source of water in pavements. Rainwater enters the road PS through cracks, infiltration from shoulders and drainage channel, high groundwater. The water gets inside the structure by an energy gradient, such as gravity, capillary forces, osmotic forces and temperature or pressure differences.

Moisture content can alter the resistance of permanent deformation in unbound granular materials dramatically [13,14]. The repeated load may cause positive pore water pressure to increase in unbound granular layers, which will consequently reduce the effective stress in unbound granular layers, which in turn decreases the stiffness and resistance to permanent deformation [15].

The permeability of the material that constitutes a road increases typically from the top of the pavement downward until about 0.7 m depth. It is estimated that about 80% of the problems encountered in pavements is related to the presence of water [16]. Otherwise, water would accumulate onto the low permeability layer and keep the upper layer wet; freezing of the accumulated water might then unbind the upper layer. Water would decrease the AC pavement bearing capacity and service life [17,18].

There is some evidence to suggest that water has less impact on the ticker and well-constructed pavements than it does on thinner ones [19]. It appears that in thicker pavements, the effect of water may be more indirect than in thinner ones, reducing material stiffness leading to later distress [17]. 

Charlie et. al [20] modeled the response of pavement at different water contents and found that an increase of moisture content from 2.3% to 4.8% led to an increase of the strains by about 60%. Chen, Sun and Yao [21] detected that base’s resilient modulus reduced significantly (approximately 30%) with each 1% increase in moisture content.

According to Salour and Erlingsson [22], collected data showed evident correlation between the moisture content and the back-calculated unbound layer stiffness. Back-calculation of the FWD deflection data revealed 63% reduction in subgrade and 48% reduction in granular layer during the spring thaw period compared to the recovered summer values. They observed a rapid increase in basin deflection during the thawing period, with a 4.7 times increase in maximum deflection value from a fully frozen structure to a thawed base and subbase layer over a 60 cm frozen subgrade.

This study aims to evaluate how temperature (of AC and subgrade layers) and moisture content (of subgrade layers) influence pavement bearing capacity. To achieve this goal, five pavement sections were selected of the road of experimental pavement structures (furthermore the Test Road) that was constructed in 2007. In this way, it was assessed the difference of the bearing capacity of the road PS among sections, loaded and unloaded lanes (right and left wheel paths and tracks). Further, the statistical analysis was done between these factors: the thickness of the AC layers, the temperature on the top of the AC surfacing and in the AC layers, and the temperature and moisture content of the soil in the subgrade and the bearing capacity. Finally, conclusions according to the results of the experimental study are represented at the end of the paper.

## 2. Main Characteristics of the Test Road

The site in Pagiriai (about 20 km from Vilnius the capital of Lithuania) was selected for constructing the Test Road. This place fulfilled all the conditions required for the experiment: it has a sufficient traffic volume of heavy vehicles, located in open terrain, has no horizontal plane curves or vertical curves in longitudinal section and could be distinguished by the same irrigation conditions within the route of the road section. The Test Road was performed on the road to the gravel quarry where the right lane is used by loaded traffic (herein loaded lane) and the left lane by unloaded traffic (herein unloaded lane; Figure 1). The total length of the Test Road is 710 m and consists of 23 sections of 30 m, one section of 20 m and three sections of 15 m in length (in total 27). The width of the Test Road carriageway is 7 m, i.e., two lanes of 3.0 m, and two shoulders of 0.5 m. The road pavement structures (PSs) are designed for 100 kN axle load ESAL_100_ = (0.8–3.0) million by Construction Technical Regulation STR 2.06.03:2001 Automobile Roads [23,24,25,26,27]. It has all necessary electronic sensors (loop profilers, temperature and moisture sensors and stress and strain transducers). The bearing capacity of subgrade was improved >100 MPa (*E*_v2_) during the construction period in each section (the bearing capacity of subgrade 459 MPa (PS No 4), 205 MPa (No 12), 150 MPa (No 18), 177 MPa (No 24) and 137 MPa (No 25)). The paper presented experimental researches in PS No 4, No 12, No 18 and No 24–25 (Table 1) because these sections are equipped with temperature and humidity sensors. The numbering of structures in this paper was accepted by the serial number of the Test Road sections.

The temperature sensors and moisture content sensors were installed in the soil of subgrade at a different depth (Table 1), and measurements were carried out continuously throughout the study period (every hour) in each section [25].

Every year in the sections, these measurements were made:Traffic monitoring;Temperature and moisture content at a different depths of the road PS;Roughness;Rutting;Longitudinal gradient and cross fall;Visual assessment of distress on the road PS;Road pavement deflection (measured with a FWD, Dynatest, Vinci, Italy and Benkelman Beam, Infratest, Brackenheim, Germany);Skid resistance (measured with a Pendulum device).

Five road PSs with various materials were selected for the study (Figure 1). AC layers of these sections were constructed from AC11WL, AC16BnL and AC32BsL. Herein AC11WL was an asphalt concrete wearing layer, with a mesh size of top sieve of 11 mm and affected by heavy loads; AC16BnL was an asphalt concrete binder layer, with a mesh size of top sieve of 16 mm and affected by heavy loads and AC32BsL was an asphalt concrete base layer, with a mesh size of top sieve of 16 mm and affected by heavy loads. Except one section, No 12, where surfacing performed from the SMA11 PMB WL (stone mastic asphalt wearing layer, with a mesh size of the top sieve of 11 mm, with polymer-modified bitumen and affected by heavy loads). The base course in all sections performed from crushed dolomite 0/56 and frost-blanket layer performed from sand 0/11. The different type of geosynthetic materials installed in sections No 24 and No 25.

## 3. Test Plan and Methods

Sixty measurements of the deflection were obtained in each section at some time. These measurements were performed after ten measurements on the left wheel path and right wheel path, including the track and repeated measurements on the loaded and unloaded lanes (Figure 2). The measurement repeated at eight times (Table 2). In total, 1920 *E*_0_ values were obtained. 

This paper presented measurements data in sections and results of a statistical analysis based on linear mixed models (LMMs). These indicators were measured and used in statistical analysis:The thickness of the asphalt concrete layer (measured by Georadar in 2015, Figures 4 and 5);The temperature on the top of the road PS (Table 1, Figure 6);The temperature at a different depth in the AC layers of the road PS (Table 1, Figure 6);Temperature and moisture content in the soil of subgrade at a different depth (Table 1, Figures 6 and 7);Surface modulus (*E*_0_, MPa) of the road PS (Figure 8);Periods (Table 2).

## 4. Statistical Analysis and Evaluation of Results

The statistical analysis (Figure 3) consisted of two parts:The first part was descriptive and contained some statistical characteristics of the factors under study (mean, median, minimum and maximum values, standard deviation and quartile) in road PS (furthermore RPS) No 4, No 12, No 18 and No 24–25. The data was visualized in various layers in box-plot diagrams;In the second part, applying LMMs [28], it was attempted to assess whether the bearing capacity of the road PS has a statistically significant difference among sections and loaded and unloaded lanes (right and left wheel paths and tracks). The effect to the bearing capacity of the road PS of these factors was analyzed: the thickness of the AC layers, the temperature on the top of the AC surfacing and in the AC layer and the temperature and moisture content of the soil in the subgrade.

### 4.1. Descriptive Statistics

On all sections, the average asphalt concrete layer was 168.80 ± 7.57 mm. A slightly bigger thickness was recorded in the road PS No 12, it was 171.90 ± 8.66 mm, while the smaller thickness was distinguished by the road PS No 4 and No 18, with a thickness of the asphalt concrete layer 167.30 ± 7.35 mm and 166.60 ± 5.91 mm, respectively (Figure 4 and Figure 5).

When comparing the thickness of the AC layer on loaded and unloaded lanes, a thicker AC layer was observed on the loaded line than in the unloaded lane, respectively 171.80 ± 7.83 mm and 165.80 ± 5.95 mm (Figure 4). When comparing the thickness of the AC layer in different lanes (Figure 5), it was seen that the right wheel path was distinguished by a wider or more varied range of AC layer thickness values and these values continued to decrease towards the left wheel path. The average thickness of the AC layer in the right wheel path was 168.30 ± 8.46 mm, in the track was 170.50 ± 7.89 mm and in the left wheel path 167.70 ± 5.82 mm. The analysis of the AC layer thickness on lanes of individual sections (Figure 5) shows that PS No 4 and No 12 of the unloaded lane had the opposite tendency, i.e., the thickness of the AC layer was observed thinner in the right wheel path of PS No 4, and the thinner AC layer was observed on the left wheel path in the PS No 12.

Figure 6 presents the data of temperature on the top of the road PS and in the AC layer at a depth of 9 cm. Results of measured temperature at a depth 8 cm and 10 cm in the AC layer did not show the results because the temperature at a depth of 8 cm, 9 cm and 10 cm varied very slightly (based on the strong correlation (above 0.9)). Therefore, the temperature at a depth of 9 cm was used to recalculate *E*_0_ values. Figure 6 presents data of temperature, and Figure 7 presents data of moisture content of soil in the subgrade of road PS in sections, respectively. On all sections, the average temperature of asphalt concrete layers was 14.40 ± 7.9 °C; the average temperature of sub-grade was 14.40 ± 5.9 °C. The average moisture content of sub-grade was 8.86% ± 4.52%.

Surface modulus *E*_0_ was calculated according to the formula that was presented together with the earlier results by Motiejūnas et al. [7]. *E*_0_ was calculated by a measured deflection that was reduced to 50 kN load and to a standard temperature of +20 °C. Bearing capacity of the asphalt layers at depths of 8, 9 and 10 cm (herein *E*_0,h8_, *E*_0,h9_ and *E*_0,h10_) were calculated by Equation (1):(1)$$E_{0,hx}=\frac{E_{0}}{10^{-0.000221\cdot h_{asf}^{1.0229}\cdot \left(T-20\right)}}$$
where *hx*—depth of asphalt layer used for calculation of *E*_0_, cm; *h*_asf_—thickness of asphalt layers, cm and *T*—a temperature of the asphalt layers measured at a certain depth, °C.

When taking measurements from all sections, the mean of *E*_0_ of the road PS was 984.40 ± 100.91 MPa, and the means of the recalculated values *E*_0,h8_, *E*_0,h9_ and *E_0_*_,h10_ were approximately the same and equal to 933.00 ± 77.87 MPa, 926.80 ± 78.34 MPa and 935.10 ± 77.92 MPa, respectively (Figure 8). 

There was no significant difference between the values of *E*_0,h8_, *E*_0,h9_ and *E*_0,h10_ because of the strong correlation (above 0.9) between the temperatures measured in AC layers at different depths (top of the road PS, 8 cm, 9 cm and 10 cm (Table 3)) and used for *E*_0_ values recalculation. Given that, only *E*_0_ and *E*_0,h9_ values were used in the analysis below.

Table 3 shows a matrix of correlations to evaluate how the factors used for analysis correlate not only with *E*_0_ but also how they relate to each other. In this correlation graph, the intensity of the color corresponds to the correlation strength, i.e., the darker blue color shows a stronger positive correlation and the darker red color shows a stronger negative correlation. Table 3 shows that *E*_0_ has a similar negative correlation (about –0.5) with all temperatures. A negative correlation means that *E*_0_ is decreasing with increasing temperature.

The comparison of *E*_0_ and *E*_0,h9_ values of the road PS in the sections (Figure 9) show that *E*_0_ and the recalculated *E*_0,h9_ values in section No 12 were on average 3−4% lower than in other sections. 

The compared *E*_0_ modulus average value in the loaded or unloaded traffic lane (Figure 10 and Figure 11) changed slightly on average 5% (on average 932–976 MPa of the loaded lane and 990–1034 MPa of the unloaded lane). It is observed that the bearing capacity of the loaded lane was lower on average up to 15% (967 MPa of the loaded lane and 1108 MPa of the unloaded lane) in a loaded lane than in an unloaded lane (Figure 10). 

When analyzing loaded and unloaded lanes of the road PS in the sections separately (Figure 11), it was seen that the road PS No 12 on the loaded lane was also characterized by a relatively lower *E*_0_ value than the other road PS on the loaded lane. However, this difference was not so high on the unloaded lane and not only the road PS No 12 but also the road PS No 18 on the unloaded lane were distinguished by somewhat lower *E*_0_ values than other sections, and the values of bearing capacity of these two sections No 12 and No 18 on the unloaded lane were very similar.

Additionally, *E*_0_ and *E*_0,h9_ values by the lanes (Figure 11) show that the right wheel path on the unloaded lane of all tested PS is distinguished by a lower *E*_0_, which is gradually increasing to the track and the left wheel path. The *E*_0_ value of the right wheel path in the road PS No 12 on the unloaded lane is the lowest compared to the other road PS. On the loaded lane, due to the lower loading all tracks of the road PS are distinguished by the lower *E*_0_, but there is no clear tendency between the left and right wheel paths on the loaded lane. The road PS No 12 again showed a significant tendency on the loaded lane. In this road PS, *E*_0_ was much lower on the loaded left wheel path than on the loaded right wheel path. There was a reverse tendency on this road PS on the unloaded lane, i.e., the lower *E*_0_ was observed on the unloaded right wheel path than on the unloaded left wheel path. Thus, it is to be seen that the road PS No 12 distinguished by its characteristics from another road PS.

During the exploitation of the road, negative factors of load are found in the path of the motor vehicle wheel. These factors influence the accelerated degradation of the road surface and the lower bearing capacity of the road PS compared to the unloaded lane, i.e., the track. The lower bearing capacity is particularly noticeable on the right wheel path of the loaded lane (up to 10%, the average *E*_0_ was 911 MPa at the right wheel path and the average *E*_0_ was 984 MPa at the left wheel path), because the right wheel path of the cross fall of the road PS was more loaded. By it, a recommendation is to install the crashed dolomite course up to the edge of the slope, thereby to strengthen with the unbound mineral mixtures the area of the shoulder.

The bearing capacity of the road PS was mainly influenced by the bearing capacity and sensitivity to the temperature and moisture content of the soils in the subgrade, the overall thickness of the road PS and the overall thickness of the AC layers with bituminous binders (Table 3). It can also be concluded that the use of fine-grained sand 0/4 for the frost-blanket layer (road PS No 4, the average *E*_0_ was 984 MPa) did not affect the bearing capacity and the temperature and moisture content of the soils in the subgrade compared to the road PS in which sand 0/11 was used (PS No 12, No 18, No 24, and No 25; the average *E*_0_ was 931 MPa). A difference of bearing capacity was about 6%. The bearing capacity of the insignificant impact is the type of unbound materials of the base course and the type of mineral mixtures with bituminous binders of base course and surfacing.

The variation of *E*_0_ and *E*_0,h9_ values were analyzed in time at each point of measurements separately in each section, i.e., in each section were 60 points where *E*_0_ was measured (Figure 12). Graphs show that the values of *E*_0_ and *E*_0,h9_ in all 60 points in each section had similar dynamics in time. The *E*_0_ was characterized by seasonal change in time, while the *E*_0,h9_ was characterized by a decrease in time, which was more noticeable at the last moments of measurement. However, these graphs also show a tendency that both *E*_0_ and *E_0_*_,h9_ values were different enough at different points in the same section. This tendency indicates that the strength of the road PS varies from one point to another.

In the second part of the statistical analysis, it was assessed what has the most significant influence on the distribution of *E*_0_ and *E*_0,h9_ values among different points of measurements, distinguishing the main factors (load, lane, right wheel path, left wheel path and track). However, the difference among points is still to be noticed, even under identical conditions. Consequently, besides the known conditions, there are specific, not observed and not included in the statistical analysis, factors, which influence the differences of *E*_0_ at different points of measurements.

Figure 12 shows that the decrease of v was observed in all the last measurements of all sections, and in section No 12, this decrease was sharper than in others. This difference of section No 12 could be influenced by a lower factor of the bearing capacity, as it was already noticed above that the average *E*_0_ of section No 12 was lower than of the other investigated sections. It can also be assumed that the significantly lower *E*_0_ values of the last measurement (April 2015) indicate that the bearing capacity of the road PS was influenced by season (hydrothermal mode). This measurement period was just a reflection of spring thaw and increased moisture content in the subgrade, but such a period did not occur during the three years of research. In this regard, it is necessary to carry out additional measurements of the bearing capacity of the hydrothermal mode in March, April and May.

Figure 13 shows the variation of *E*_0_ and *E*_0,h9_ values in each of the four sections, excluding the dynamics of each point separately. The tendency of *E*_0_ and *E*_0,h9_ were similar in all tested PS. The seasonal increase and decrease in the *E*_0_ values over time were observed. It can be assumed that this seasonal impact in time was influenced by change of the temperature and moisture content in the soil of the subgrade during the year. When comparing the values of *E*_0_ and *E*_0,h9_, it was observed that even after recalculating *E*_0_ to *E*_0,h9_ it was not always possible to avoid seasonal change in the data. The lowest average *E*_0_ value measured in July 2014 was 871 MPa (air temperature was 25 °C, Table 2) and the highest was 1112 MPa in November 2013 (air temperature was 5 °C, Table 2) and the lowest average *E*_0,h9_ value measured in April 2015 was 834 MPa (air temperature was 9 °C, Table 2) and the highest was 969 MPa in November 2013 (air temperature was 5 °C Table 2; Figure 14 and Figure 15). That also reflects in the road PS No 12, where the deviation from the mean of the *E*_0,h9_ values was almost 5% in July 2014 and 8% in April 2015. At the same time of measurement, the similar deviations from the mean were observed in the road PS No 4, the deviations from the mean were 3% and 9% (Figure 14 and Figure 15), respectively. The type of improvement of the subgrade could influence this difference because in different sections the soil of the subgrade was improved with appropriately used asphalt granules or mineral materials of different fractions.

By analyzing the data of Figure 12 and Figure 13, both the dynamics of individual values of *E*_0_ and *E*_0,h9_, as well as the mean values, it was found that by neutralizing the influence of asphalt temperature on the bearing capacity of the road PS, the effect of seasonal impact on the weakening of the subgrade remained. Due to the lower ambient temperature, the more strength asphalt in the autumn and spring seasons was not sufficient to compensate for the decrease in the bearing capacity of the subgrade due to the increased moisture content of the soil in the subgrade. Even a subgrade of no less than 100 MPa, when such a bearing capacity was only achieved by improving the soil in the subgrade, did not ensure the complete elimination of seasonal effects. Therefore, only reinforcement of the soil in the subgrade should be applied in the road PS of Class I (where ESAL_100_ = (10.0–32.0) million) and Class SV (where ESAL_100_ is more than 32.0 million), and the improvement or reinforcement of the soil in the subgrade should be applied in the road PS of Class III (where ESAL_100_ = (0.8–3.0) million) and Class II (where ESAL_100_ = (3.0–10.0) million). Analyzing data more in details was decided that this is a seasonal effect on the change in bearing capacity of the road PS.

The values of *E*_0_ and *E*_0,h9_ of all five road PS in sections measured at different measuring periods were combined to conclude the entire Test Road. Therefore, an analysis in a time of the variation of *E*_0_ and *E*_0,h9_ values together was carried out (Figure 16). According to Figure 16, it was seen that the same tendency as in the analysis of data for each section separately, *E*_0_ and *E*_0,h9_ mean values changed over time (increasing and decreasing) without seasonal adjustment.

By analyzing the dynamics of *E*_0_ and *E*_0,h9_ values in each individual point (left wheel path, right wheel path and track) on loaded and unloaded lanes of each section were found that in the loaded lane, the dispersion of the measuring values of individual points was higher than in the unloaded lane (Figure 17, Figure 18 and Figure 19). In unloaded lanes, the *E*_0_ values at each point had small dispersion, i.e., the curves representing the *E*_0_ dynamics at each point of the PS were somewhat flattened, while in loaded lanes at different points of the same section *E*_0_ was entirely different, i.e., the curves depicting the *E*_0_ dynamics at each point of the road PS seemed to be separated from each other.

Therefore, it was assumed that in the loaded lane, the values of *E*_0_ and *E*_0,h9_ differed at individual points due to the uneven loading from the wheels of the motor vehicle. The uneven load was due to the movement of the motor vehicle wheel along the driving trajectory. As well as during the measurements, the FWD might have a partial deviation in time from the point of measurement, although the latter assumption was denied by the glare of the measured values in the unloaded lane.

Observed differences in distribution patterns between measured data on loaded and unloaded lanes were further analyzed at a more aggregated level. The graphs of Figure 19 show the dynamics of the *E*_0_ and *E*_0,h9_ values at each measuring point by analyzing the data of all tested PS together but separating loaded and unloaded lanes. It can be seen that the values of the strength of the road PS in the unloaded lanes were higher and with less dispersion than in loaded lanes. From the given dynamics *E*_0,h9_ on different lanes, it could also be seen that the right wheel path of the unloaded lane had a more pronounced downward tendency of *E*_0,h9_ than the left wheel path or track. It could be suspected that in the right wheel path of the unloaded lane the strength of the road PS for some reason decreased faster. In the loaded lane, it was hard to see differences, as the point dynamics were quite chaotic.

Therefore, the effect of heavy vehicle loads had the most effect on the path trajectory of the right wheel, otherwise defined, on the right wheel path, which was closer to the shoulder. In this regard, it was recommended that during the construction or reconstruction of the road PS of Class III (where ESAL_100_ = (0.8–3.0) million) and a higher class, the width of the crushed dolomite course should be extended to the slope. Thus ensuring the stability and bearing capacity of the edge of the road and the layers of the road PS in the area of the shoulder.

The bearing capacity during measuring autumn and spring periods of the loaded lane (Figure 20) changed respectively up to 6.3% (in autumn) and up to 8.3% (in spring; in PS No 4), up to 4.9% and up to 8.7% (No 12), up to 0.8% and up to 2.1% (No 18) and up to 1.9% and up to 5.3% (No 24–25). The research results of the bearing capacity of the unloaded lane (Figure 19) changed respectively up to 0.5% (in autumn) and up to 12.5% (in spring; No 4), up to 5.5% and up to 9.3% (No 12), up to 2.2% and up to 1.5% (No 18) and up to 4.8% and up to 7.9% (No 24–25). The temperature of asphalt pavement surface was on average about 14–18 °C (in autumn) and 3–10 °C (in spring; Table 2). The bearing capacity results of summer measurement (when temperature of asphalt pavement surface was in average about 20–31 °C, Table 2) were on average up to 12.5% lower than in autumn and up to 6.1% lower than in spring (Figure 20 and Figure 21).

### 4.2. The Effect of the Thickness of the Asphalt Concrete Layers for the E_0_ and E_0,h9_


The higher total thickness of the AC layers in the loaded lane than in the unloaded lane, along with other factors, affected the bearing capacity of the tested PS, i.e., *E*_0_ numerical values. The numerical values of *E*_0_ varied from 757 to 1287 MPa (average 1012.00 ± 96.97 MPa) in the unloaded lane and from 870 to 1228 MPa (average 956.50 ± 97.10 MPa) in the loaded lane. The numerical values of *E*_0,h9_ varied from 703.60 to 1134.00 MPa (average 952.40 ± 69.51 MPa) in the unloaded lane and from 606 to 1135 MPa (average 901.20 ± 78.32 MPa) in the loaded lane.

By analyzing the recalculated values *E*_0,h9_, there was a significantly lower distribution of these values than the *E*_0_ (Figure 22 and Figure 23). The tendency that the mean value *E*_i_ of the PS in an unloaded lane in which the total average thickness of the AC layers was 165.80 ± 5.95 mm was more than the loaded lane in which the total average thickness of the AC layers was about 171.80 ± 7.83 mm that remained.

It has been observed in studies that, regardless of the overall thickness of the AC layers, the average value *E*_i_ of the tested PS in the loaded or unloaded lanes varies slightly, but when compared to the average values *E*_i_ of the loaded and unloaded PS, this tendency is not valid. In this way, it was concluded that the total heavy vehicle loads had a significant effect on the *E* of the tested PS.

Figure 24 shows that the temperatures and dispersion graphs of *E*_0_ and *E*_0,h9_ indicate that there was an *E*_0_ linear negative temperature dependence at a given depth (the higher the temperature, the lower the *E*_0_).

During the study, the average moisture content at a depth of 100 cm was 6.344% ± 3.68%, at a depth of 130 cm, 8.36% ± 4.21% and at a depth of 150 cm, 11.88% ± 3.67%. From moisture content at a given depth, *E*_0_ and at a given depth of moisture content and the *E*_0,h9_ dispersion graph (Figure 25), it can be seen that neither *E*_0_ nor *E*_0,h9_ were related to moisture content.

### 4.3. Fitting Linear Mixed Models (LMMs)

In this statistical analysis, applying LMMs to clustered data, in which the units of analysis (points in the road) were within clusters (sample of four different road sections), and repeated measures in these different points were collected over time with unequally spaced time intervals. The data were balanced—each cluster had the same number of units, which were measured at the same number of time points. Points in the section of the road PS were dependent, as this dependence was due to the different and specific characteristics of the particular road PS. Due to the data and different clusters in which data could be correlated, linear regression models (LRMs) were not appropriate here, and therefore LMMs were applied. LMMs, like LRMs, are linear in the parameters, but independent variables in LMMs involve not only fixed but also random effects [28]. Fixed effects are treated the same as regression coefficients in LMMs—they indicate the relationships of the covariates with the continuous outcome variable. Random effects are used in modeling the random variation in the dependent variable at different levels of the data. This random variation occurs because each point in the section of the road PS is very individual and examining even the same characteristics of these points; these characteristics depend not only on the features included in the research but also on the individual, unknown peculiarities of the researchers. Thus, the expression of the LMM for the *i*^th^ subject (n is the number of all survey subjects), whose measurements of the test variable values are repeated *k* times, is [30] (Equations (2)–(3)):(2)$$\begin{array}{c} Y_{i}=X_{i}\;\beta +Z_{i}\;u_{i}+\varepsilon_{i},\;\;\;i=1,\ldots ,n,\\ u_{i}\sim N(0,\;D\;),\;\;\;\varepsilon_{i}\sim N(0,\;R_{i}\;), \end{array}$$
(3)$$Y_{i}=\left(\begin{array}{c} Y_{1i}\\ Y_{2i}\\ \ldots \\ Y_{ki} \end{array}\right),\;X_{i}=\left(\begin{array}{c} X_{1i}^{\left(1\right)}\\ X_{2i}^{\left(1\right)}\\ \ldots \\ X_{ki}^{\left(1\right)} \end{array}\begin{array}{c} \ldots \\ \ldots \\ \ldots \\ \ldots \end{array}\;\;\begin{array}{c} X_{1i}^{\left(p\right)}\\ X_{2i}^{\left(p\right)}\\ \ldots \\ X_{ki}^{\left(p\right)} \end{array}\right),\;\beta =\left(\begin{array}{c} \beta_{1}\\ \beta_{2}\\ \ldots \\ \beta_{p} \end{array}\right),\;Z_{i}=\left(\begin{array}{c} Z_{1i}^{\left(1\right)}\\ Z_{2i}^{\left(1\right)}\\ \ldots \\ Z_{ki}^{\left(1\right)} \end{array}\begin{array}{c} \ldots \\ \ldots \\ \ldots \\ \ldots \end{array}\;\;\begin{array}{c} Z_{1i}^{\left(q\right)}\\ Z_{2i}^{\left(q\right)}\\ \ldots \\ Z_{ki}^{\left(q\right)} \end{array}\right),\;u_{i}=\left(\begin{array}{c} u_{1i}\\ u_{2i}\\ \ldots \\ u_{qi} \end{array}\right),\;\varepsilon_{i}=\left(\begin{array}{c} \varepsilon_{1i}\\ \varepsilon_{2i}\\ \ldots \\ \varepsilon_{ki} \end{array}\right),$$
where *Y_i_*—dependent variable. The dependent variable in this study is the surface modulus *E*_0_ or, according to the method proposed by previous researchers, recalculated *E*_0,h9_; 

*X_i_*—a matrix of explanatory variables in which the number of variables p corresponds to the number of fixed parameters and their interactions, included in the model. The explanatory variables of this study were of two types: (1) road section, lane, factor of the load on the road PS and thickness of the AC layers are indications whose values are unchanging over time; and (2) the temperature of different types and moisture content, as well as the values of the dependent variable, vary depending on weather conditions, during a particular measurement (Table 4); 

*Z_i_*—matrix of random effects *q*. For example, if the model contains only two random parameters, one of which is for estimating the randomness of the intercept and the other one for evaluating the randomness of slope between the subjects, then the random effects matrix *Z_i_* consists of two columns, i.e., the unit column and column of the time moments, respectively: $Z_{i}=\left(\begin{array}{c} 1\\ 1\\ \ldots \\ 1 \end{array}\begin{array}{c} \ldots \\ \ldots \\ \ldots \\ \ldots \end{array}\;\;\begin{array}{c} 1\\ 2\\ \ldots \\ k \end{array}\right)$,

with the vector of parameters $u_{i}=\left(\begin{array}{c} u_{1i}\\ u_{2i} \end{array}\right)$.

Assuming that the mean of the random parameters is zero, the random variables are estimated with a covariance matrix $D=\left(\begin{array}{cc} Var(u_{1i}) & \mathrm{cov}(u_{1i},u_{2i})\\ \mathrm{cov}(u_{2i},u_{1i}) & Var(u_{2i}) \end{array}\right)$, whose diagonal elements correspond to the variance of the parameters $u_{1i}$ and $u_{2i}$, and non-diagonal elements correspond to the covariance among these random parameters. 

*ε_i_*—model residuals, where the covariance matrix *R_i_* defines residuals dependence due to repeated measurements. This matrix could be diagonal, $R_{i}=Var(\varepsilon_{i})=\left(\begin{array}{ccc} \sigma^{2} & & \\ & \ldots & \\ & & \sigma^{2} \end{array}\right),\;\;k\times k$, which shows the dispersion of residuals of the ith subject at each measurement moment, but it can also be of a different structure [30].

LMMs are implemented in many software packages. Appropriate procedures could be used in most popular statistical packages: SAS, SPSS and R [28]. This statistical analysis used the R program package and its library *nlme* [31].

This study used a top-down LMM model specification strategy [28]. First of all the model with all fixed parameters were estimated (time, lane, load and interactions of these variables), in the second step possible random effects were included in the model and finally appropriate covariance matrix structure for model residuals were estimated. To evaluate which of the two models was more appropriate the Likelihood ratio test was used [28].

#### 4.3.1. M1 Model for Dependent Variable E_0,h9_

In the model M1, *E*_0,h9_ was the dependent variable, whose values were recalculated from *E*_0_ removing the effect of the recent temperature on the AC layer at a depth of 9 cm and taking into account the thickness of the AC layer. This correction of the bearing capacity should smooth the dynamics of the initial *E*_0_ values in time, eliminating the apparent seasonal impact. However, Table 3 shows that seasonal change had not been successfully removed in all cases. Therefore, it was assumed that *E*_0_ values depended not only on the temperature of the AC layer but also on other seasonal factors that were not included in the study. Another reason might be that at a specific temperature, the bearing capacity was not very typical for the formula used for recalculation. Therefore, a slight seasonal impact was seen in the recalculated data.

Since *E*_0,h9_ had a slightly different level among the different pavement sections (Figure 12), the random intercept parameter between the pavement sections in the M1 was included. Figure 12 shows that in all of the studied pavement sections there was a decrease in the *E*_0,h9_ values in time, but this did not happen equally quickly across all sections (the slope was different). Due to this characteristic, the random trend among the sections was included in model M1. By taking all the points in each pavement section, their *E*_0,h9_ values were also scattered widely, and this could lead to the assumption that there was a certain randomness among the points that were not described by the fixed parameters. To estimate this randomness random intercept for individual points in the sections was included in model M1. Since the dynamics of *E*_0,h9_ in single points on individual sections were quite similar in time, the random trend of single points in the section was not included in M1.

In the final M1 model fixed variables (lane, load and time) and the random effects associated to pavement sections (clusters) and pavement points in road sections were included. To estimate residuals dependence in the model a diagonal covariance matrix with different dispersions of residuals at each time point was chosen The final model for *E*_0,h9_ in the i-point (i = 1, …, 60) on the j-section (j = 1, …, 4) at a given time t (t = 1, …, 8) had this form [29] Equation (4):(4)$$\begin{array}{l} E_{0,h9}{}_{{}_{tij}}=\beta_{0}+\beta_{1}L_{t}+\beta_{2}K_{ij}+\beta_{3}T_{ij}+\beta_{4}A_{ij}+\beta_{5}L_{t}\times A_{ij}+\beta_{6}L_{t}\times K_{ij}+\beta_{7}L_{t}\times T_{ij}+\\ +u_{0_{j}}+u_{1_{j}}\times L+u_{0_{ij}}+\varepsilon_{tij} \end{array},$$
where *L*—time (measurement at a given time), *K*—indicator variable indicating the left wheel path; *T*—indicator variable indicating the track, *A*—indicator variable indicating loaded lane, *L × A*—interaction of time and load variables, which estimates the differences between the dynamics of the bearing capacity of the road PS on loaded and unloaded lanes; *L × K*—interaction of time and left wheel path variables, which estimates the differences between the dynamics of the bearing capacity of the road PS in the left and right wheel paths; *L × T*—interaction of the time and track variables, which estimates the differences between the dynamics of the bearing capacity of the road PS in the track and right wheel paths; *β_0_ – β_7_*—parameters that correspond to fixed effects in the model, i.e., intercept, time, lane, load and interactions of these effects; $u_{0_{ij}}$ is a random effect associated to the difference of the measured points in the separate section of the road PS and $\varepsilon_{ij}$ is the model residuals. 

The random parameters $u_{0_{j}}$ and $u_{1_{j}}$ describe the remaining differences of *E*_0,h9_, which have not been estimated by the fixed parameters in the M1 model. These random parameters correspond to the differences between the intercept and the direction of the trend between the sections. The mean of these random parameters is zero, and their variance and covariance are given in the *D* matrix. In this matrix, the diagonal elements correspond to the dispersion of the random effects $u_{0_{j}}$ and $u_{1_{j}}$. Non-diagonal elements are the covariance among these random parameters. This model assumes that the vector $\varepsilon_{\mathrm{ij}}$ for a given subject is a random variable that follows a multivariate normal distribution with a zero mean and a diagonal dispersion matrix $R_{ij}$ (Equation (5)):(5)$$\varepsilon_{ij}=\left(\begin{array}{c} \varepsilon_{1ij}\\ \ldots \\ \varepsilon_{8ij} \end{array}\right)\sim N(0,\;R_{ij}),\;\;R_{ij}=\left(\begin{array}{cccc} r_{11} & 0 & 0 & 0\\ 0 & r_{22} & 0 & 0\\ 0 & 0 & \ldots & 0\\ 0 & 0 & 0 & r_{88} \end{array}\right).$$

All fixed parameters were statistically significant in the M1 model (Table 5). Negative parameters, which are related to time and load variables, show correspondingly the significant decrease of the bearing capacity of the road PS in time and the negative influence of load on the bearing capacity. At loaded points *E*_0,h9_ was an average of 88.5 MPa lower than unloaded points. The dynamics of the bearing capacity of loaded and unloaded points varied in time differently, i.e., at loaded points bearing capacity values decreased slower than the at unloaded points (parameter *β_5_*; Table 5). Eliminating the time effects, there were still significant differences of *E*_0,h9_ between different ruts: The bearing capacity of the road PS on the left wheel path and the track was less than on the right wheel path.

#### 4.3.2. M2 Model for Dependent Variable *E_0_*

This model estimates the same linear mixed model as M1 (with the same fixed and random parameters), but the dependent variable in it is *E*_0_ values from which are not eliminated the influence of temperature and thickness of the AC layers. Therefore additional fixed parameters related to the temperature, thickness of the AC layers and interaction between temperature and moisture content were included in the M2 model respectively: *β_8_*, *β_9_* and *β1_0_* parameters. The negative correlation between *E*_0_ and different type of temperatures was observed. Since the various types of temperature strongly correlated with each other (correlation >0.9), only the temperature in the AC layer at a depth of 9 cm was included in the M2 model. This temperature was used to calibrate the *E*_0,h9_ values that were used in the M1 model (Equation (2)). Thus, the expression for model M2 is (Equation (6)):(6)$$E_{\begin{array}{c} 0_{tij}=\beta_{0}+\beta_{1}L_{t}+\beta_{2}K_{ij}+\beta_{3}T_{ij}+\beta_{4}A_{ij}+\beta_{5}L_{t}\times A_{ij}+\beta_{6}L_{t}\times K_{ij}+\beta_{7}L_{t}\times T_{ij}+\\ \beta_{8}TEMP+\beta_{9}ASPH+\beta_{10}TEMP\times MOISTURE+u_{0j}+u_{1j}\times L+u_{0_{ij}+\varepsilon_{tij}.} \end{array}}$$

Almost all fixed parameters in the M2 model were statistically significant with similar parameters estimates as in the M1 model (Table 5). The additional variables in the M2 model, temperature and the thickness of the AC layer had accordingly the negative, and the positive effect on the bearing capacity of the road pavement construction. When the values of all other parameters in the model were fixed, then increasing the temperature by 1 °C at a depth of 9 cm in the AC layer influenced *E*_0_ decreases by an average of 8.41 MPa, and increasing thickness of the AC layer by 10 mm, *E*_0_ increases by an average of almost 44 MPa. The interaction of the temperature and the moisture content was statistically non-significant (*p* = 0.391) in the M2 model.

The estimation of random variables parameters (dispersion and covariance) of the M1 and M2 models are also presented in Table 5. It can be seen that the random parameters of a section were interdependent; their correlation was −0.594 in the M1 model and −0.34 in the M2 model. Therefore, if a higher intercept is estimated in a particular section, then there is a lower slope in this section estimated. In other words, if the bearing capacity of a road PS is higher in a particular section, then values in time decrease slower (or vice versa. In this case, the correlation does not indicate causality). Statistically significant random parameters between measured points in each section could be related to different loading rate of each measured point in the section, which is not included in the survey data. It is known only that the point on the road pavement is loaded or not, but how much it is loaded is not known. 

The residual values of the models presented in Figure 26 could be used to determine how adequate the fitted models are. Figure 26 shows that the residual values of the M1 model were more concentrated than the M2 model. Such dispersion might be affected by the fact that for the M1 model recalculated *E*_0,h9_ values, with a smaller dispersion than initial *E*_0_ values were used.

On the other hand, at this stage, the purpose of the LMM models was not the prediction of the strength of the road PS, but the assessment of the strength differences between the different places in the road PS and properties of the road PS. Further research needs to include additional variables that describe the road PS to predict the strength of the road PS.

## 5. Conclusions

The possibilities of the LMM model allowed evaluating the dependence of the bearing capacity of the road PS on various fixed characteristics: the lane (unloaded or loaded) and their load mode, and these interactions, by finding specific random differences between points. These differences were not explained by the fixed characteristics included in the model. One of the reasons was that, even with the evaluation of fixed parameters, the bearing capacity of the road PS was different at points, there might be certain factors not included in the survey, i.e., structural characteristics of the section (cluster) or load size for each point at section (cluster). In this analysis, the load characteristic had only two values, i.e., loaded and unloaded. It was not known how much each point was loaded.Heavy vehicle loads affected the performance of road pavement and this effect was more significant (aggressive) in the trajectory of the path of the right wheels. Due to the lower bearing capacity of roadside up to 10% (the average *E*_0_ was 911 MPa at the right wheel path, and the average *E*_0_ was 984 MPa at the left wheel path) and ending of unbound base layer at the edge of asphalt and higher wheel stress asphalt wearing and binder layers rutted much more in right wheel path. Considering this, it was recommended for roads with ESAL’s of 1.0 mln and higher to prolong unbound base layer until slope of the roadside; this would increase structural strength of PS and decrease the depth of wheel stress.The usage of fine-grained sand 0/4 (the average *E*_0_ was 984 MPa) for the frost-blanket layer did not influence the PS bearing capacity and hydrothermal conditions, comparing with PS with the frost-blanket layer of sand 0/11 (the average *E*_0_ was 984 MPa). A difference of the bearing capacity was about 6%.Independently from asphalt layers overall thickness, *E*_0_ modulus average value in loaded or unloaded traffic lane changed slightly on average 5% (on average 932–976 MPa of the loaded lane and 990–1034 MPa of the unloaded lane). However, comparing the *E*_0_ modulus average values of loaded and unloaded traffic lanes, this tendency was not valid (the difference was up to 15%, 967 MPa of loaded lane and 1108 MPa of the unloaded lane). This way it was concluded that total traffic loads had a significant impact to the E modulus of PS.The most significant influence for pavement structural strength had the bearing capacity of the subgrade, susceptibility to hydrothermal impact, the thickness of asphalt layers and thickness of the whole PS. The unbound and bound layers only slightly influenced pavement structural strength.Analysis of overall *E*_0_ and *E*_0,h9_ data declared that seasonal impact on pavement structural strength due to a change of subgrade bearing capacity remained after correction of asphalt stiffness dependent on layer temperature. This assigned that stiffer asphalt layers in autumn and springtime were insufficient to compensate for the decrease of pavement structural strength due to decreased bearing capacity of subgrade. The bearing capacity during measuring autumn and spring periods of loaded lane changed respectively up to 6.3% (in autumn) and up to 8.3% (in spring; in PS No 4), up to 4.9% and up to 8.7% (No 12), up to 0.8% and up to 2.1% (No 18) and up to 1.9% and up to 5.3% (No 24–25). The temperature of asphalt pavement surface was on average about 14–18 °C (in autumn) and 3–10 °C (in spring). The bearing capacity results of the summer measurement (when temperature of asphalt pavement surface was on average about 20–31 °C) were one average up to 12.5% lower than in autumn and up to 6.1% lower than in spring.A subgrade improvement with the aggregates mix reaching deformation modulus EV2 higher than 100 MPa did not ensure the complete elimination of seasonal impact. Due to this, it was recommended for roads with ESAL’s of 3.0 mln and higher to use subgrade stabilization and for roads with ESAL’s of lower than 3.0 mln to use subgrade improvement or stabilization.It was detected that neither *E*_0_ nor *E*_0,h9_ were related to moisture content at a depths of 100 cm, 130 cm and 150 cm.This analysis was an initial step towards more detailed testing of road PS in the Test Road, it found some of the characteristics of the road PS and allowed for the adoption of appropriate engineering solutions.

## Figures and Tables

**Figure 1 materials-12-03826-f001:**
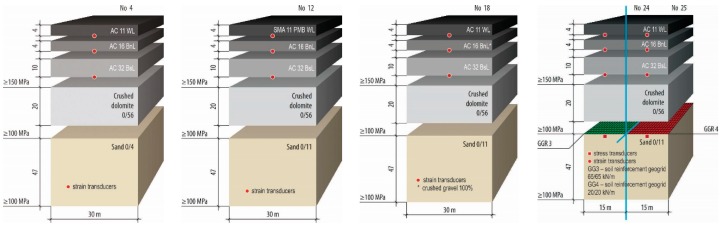
PSs of the Test Road: No 4, No 12, No 18 and No 24–25.

**Figure 2 materials-12-03826-f002:**
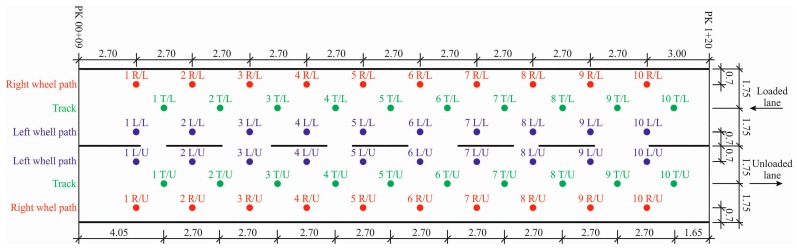
The layout of points of measured deflection in the tested PS No 4 (where: R/L—right wheel path of the loaded lane, T/L—track of the loaded lane, L/L—left wheel path of the loaded lane, R/U—right wheel path of the unloaded lane, T/U—track of the unloaded lane and L/U—left wheel path of the unloaded lane. Distances between measuring points are given in meters (m)

**Figure 3 materials-12-03826-f003:**
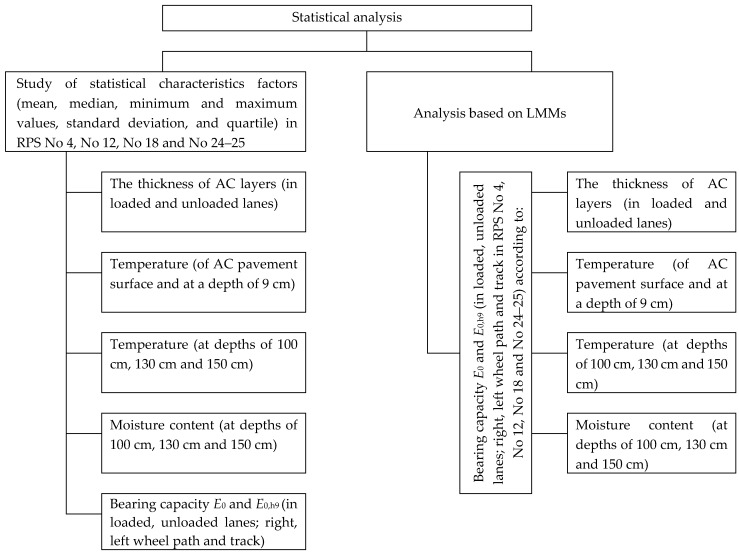
Statistical analysis of the experiment.

**Figure 4 materials-12-03826-f004:**
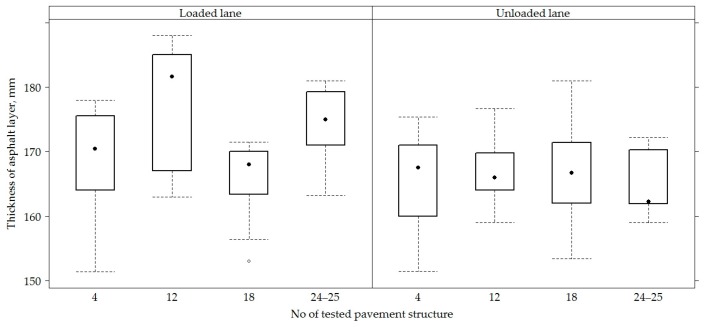
The box-plot of the total thickness of asphalt concrete layers in loaded and unloaded lanes.

**Figure 5 materials-12-03826-f005:**
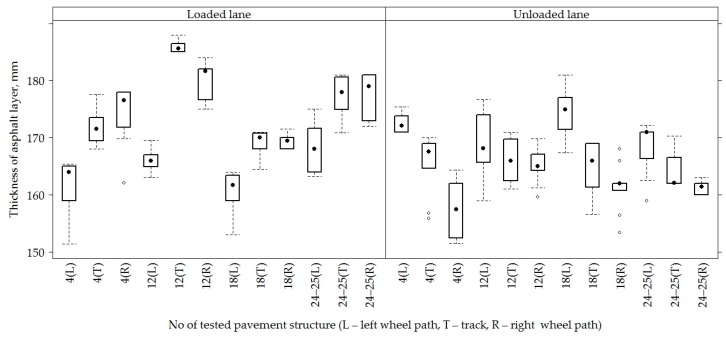
The box-plot of the total thickness of the asphalt concrete (AC) layers in different PSs on different lanes.

**Figure 6 materials-12-03826-f006:**
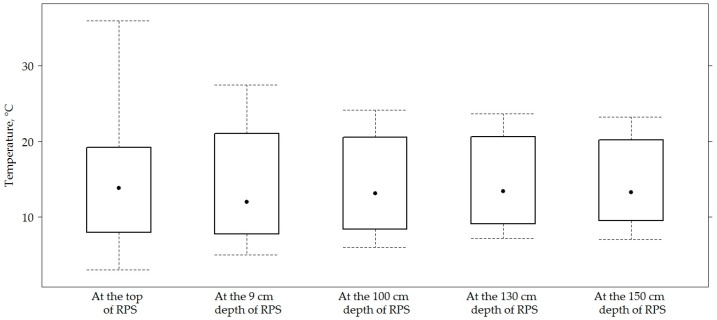
The box-plot of the measured temperature.

**Figure 7 materials-12-03826-f007:**
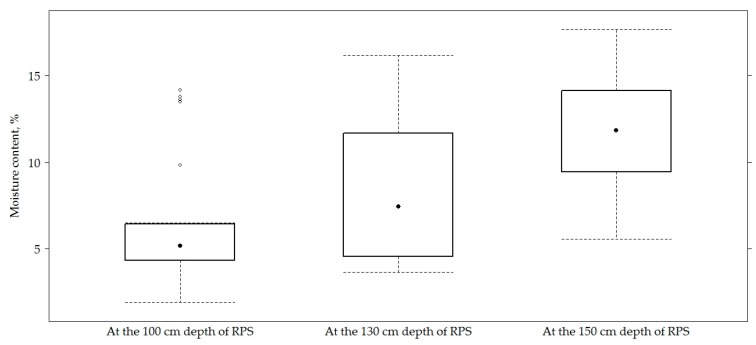
The box-plot of moisture content of the soils in the subgrade.

**Figure 8 materials-12-03826-f008:**
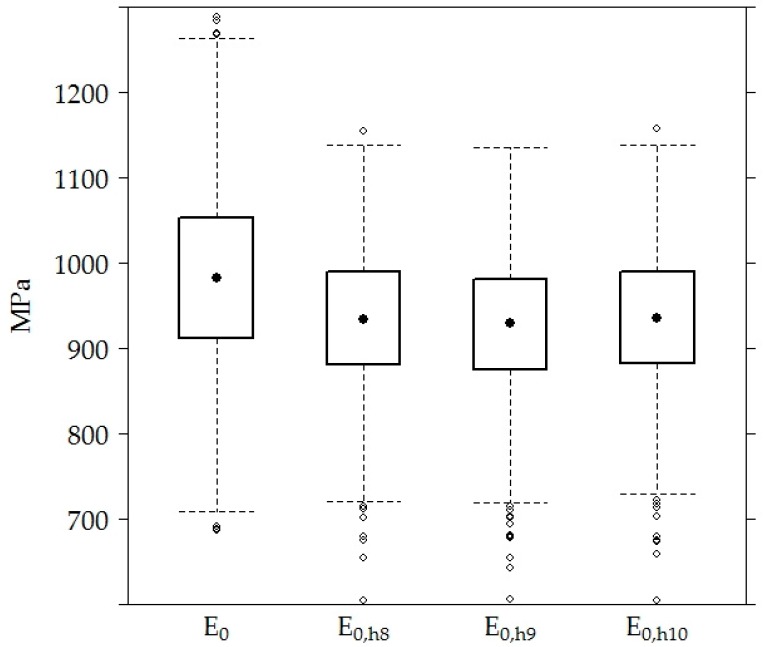
The box-plot of the surface and separate layer modulus of the road PS.

**Figure 9 materials-12-03826-f009:**
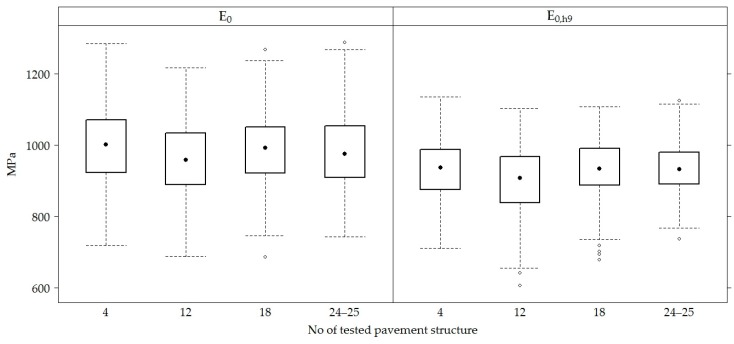
The box-plots of *E*_0_ and *E*_0,h9_ values in each tested road section.

**Figure 10 materials-12-03826-f010:**
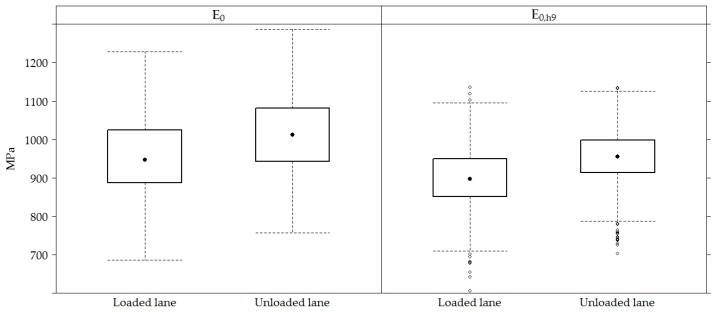
The box-plots of *E*_0_ and *E*_0,h9_ values in loaded and unloaded lanes.

**Figure 11 materials-12-03826-f011:**
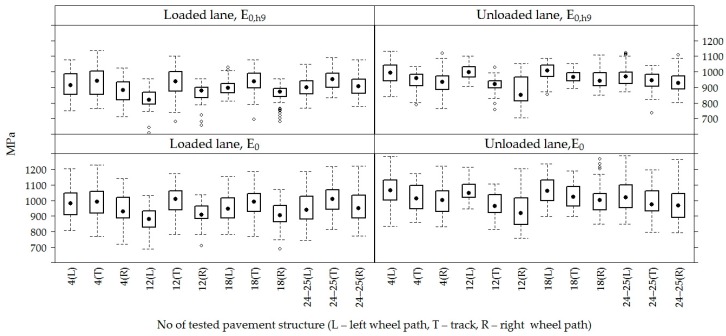
The box-plots of *E*_0_ and *E*_0,h9_ values in each loaded and unloaded lane by distinguishing the left wheel path, track and right wheel path.

**Figure 12 materials-12-03826-f012:**
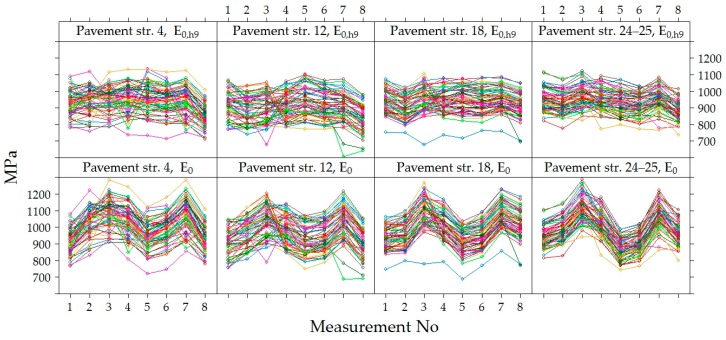
*E*_0_ and *E*_0,h9_ at individual measuring points [29].

**Figure 13 materials-12-03826-f013:**
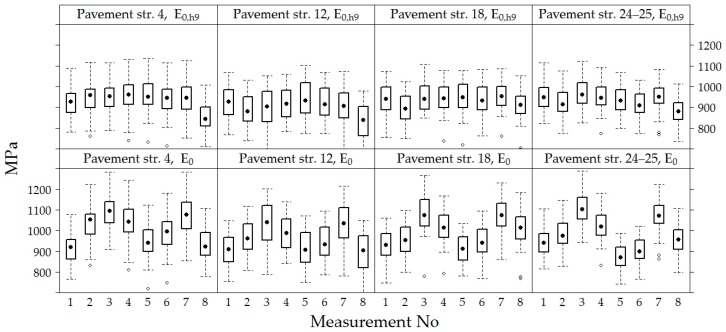
The distribution of individual *E*_0_ and *E*_0,h9_ values for each of the road PS.

**Figure 14 materials-12-03826-f014:**
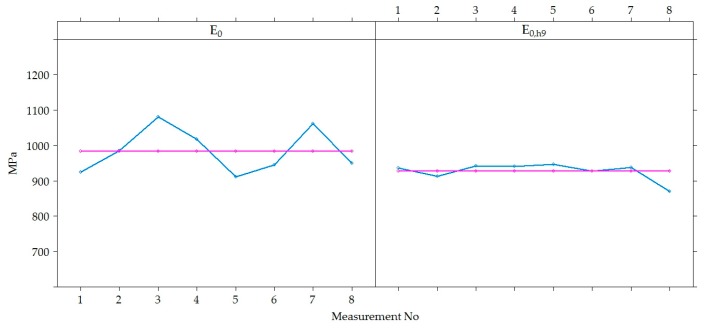
*E*_0_ and *E*_0,h9_ mean values and their fluctuations regarding the total average over time.

**Figure 15 materials-12-03826-f015:**
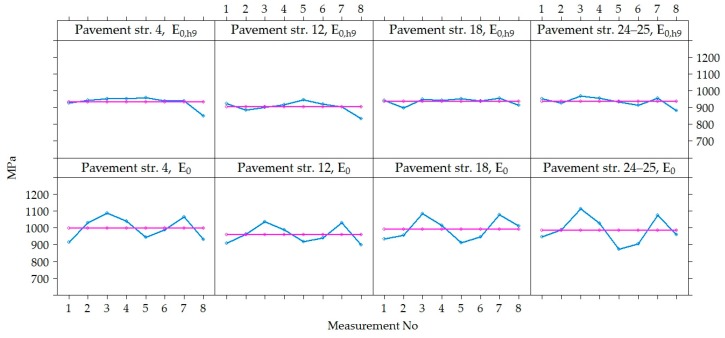
*E*_0_ and *E*_0,h9_ mean values and their fluctuations regarding the total average in each section.

**Figure 16 materials-12-03826-f016:**
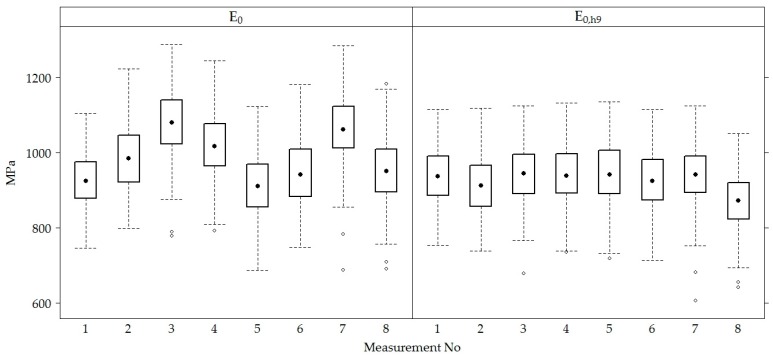
The box-plots of the distribution of individual *E*_0_ and *E*_0,h9_ values of the tested PS.

**Figure 17 materials-12-03826-f017:**
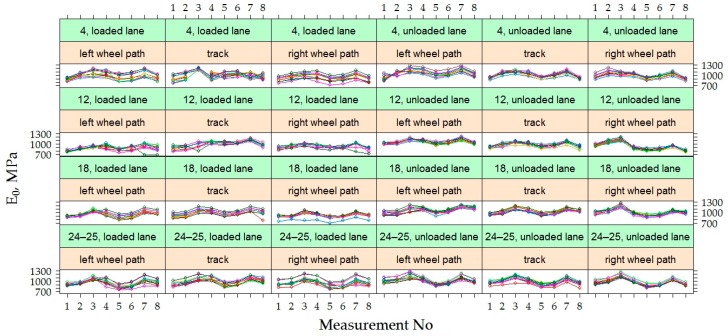
The dynamic of the individual *E*_0_ values of each section for each point of measurement on the right and left wheel paths and in the loaded and unloaded lanes.

**Figure 18 materials-12-03826-f018:**
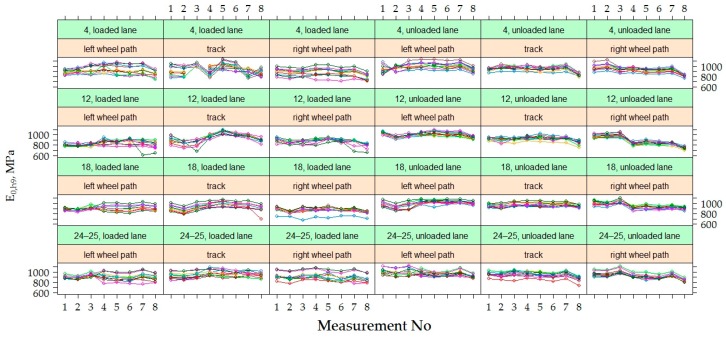
The dynamic of the individual *E*_0,h9_ values of each section for each point of measurement on the right and left wheel paths and in the loaded and unloaded lanes.

**Figure 19 materials-12-03826-f019:**
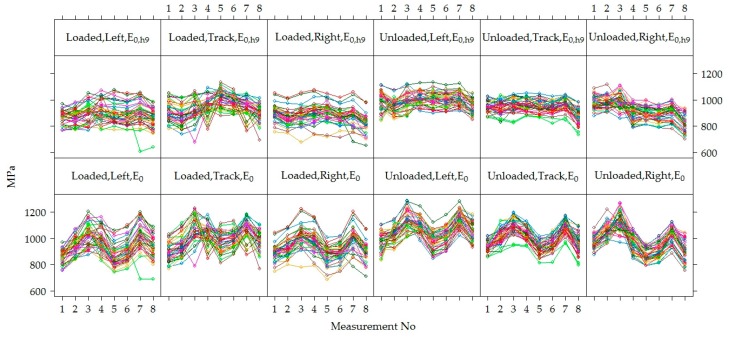
The dynamics of *E*_0_ and *E*_0,h9_ values of each point of measurement in unloaded and loaded lanes by analyzing the data of all road PS together.

**Figure 20 materials-12-03826-f020:**
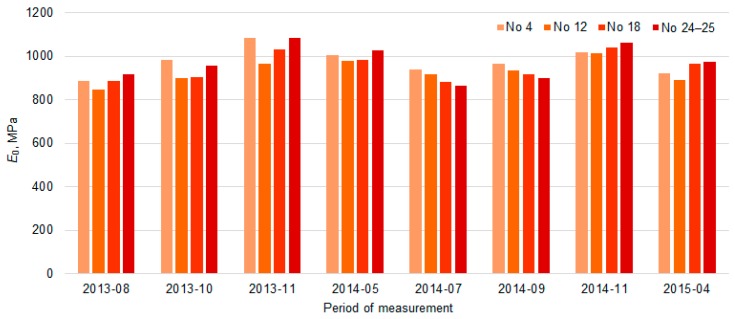
The average values of bearing capacity in different periods of the loaded lane of PS No 4, No 12, No 18 and No 24–25.

**Figure 21 materials-12-03826-f021:**
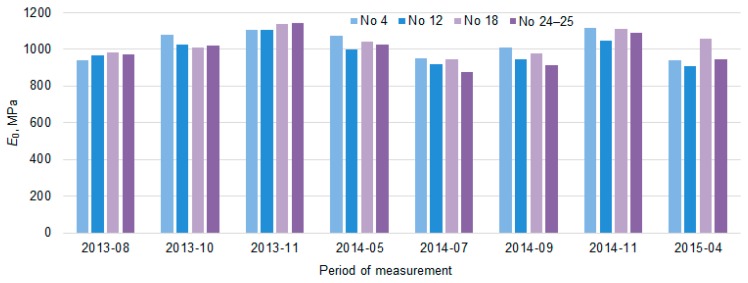
The average values of bearing capacity in different periods of the unloaded lane of PS No 4, No 12, No 18 and No 24–25.

**Figure 22 materials-12-03826-f022:**
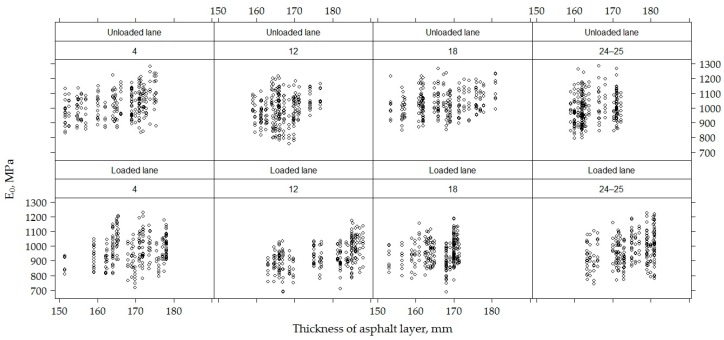
Influence of the total thickness of asphalt concrete layers on *E*_0_ by analyzing each section individually in loaded and unloaded lanes.

**Figure 23 materials-12-03826-f023:**
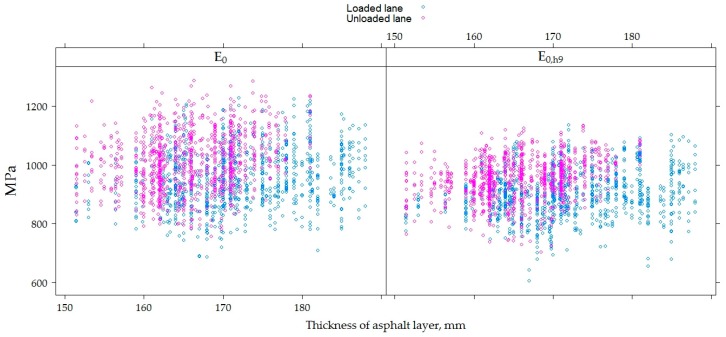
Influence of total thickness of asphalt concrete layers on *E*_0_ and *E*_0,h9_ values in loaded and unloaded lanes.

**Figure 24 materials-12-03826-f024:**
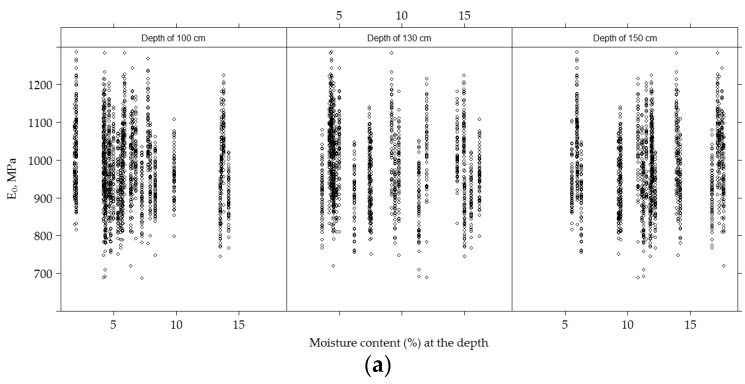
Scatter plots for *E*_0_ (**a**) and *E*_0,h9_ (**b**) values and moisture content at a given depth.

**Figure 25 materials-12-03826-f025:**
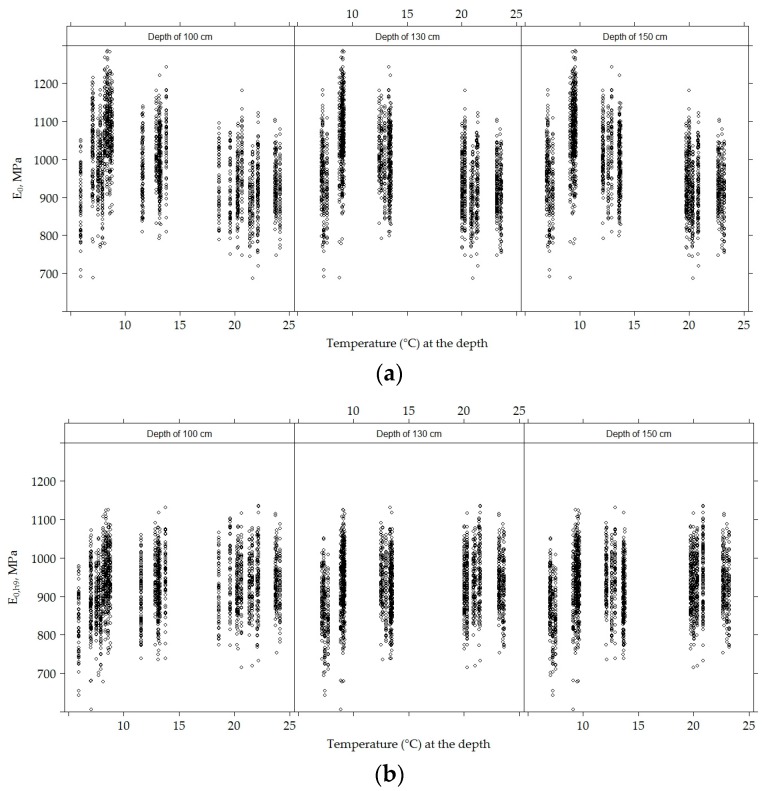
Scatter plots for *E*_0_ (**a**) and *E*_0,h9_ (**b**) values and temperature content at a given depth.

**Figure 26 materials-12-03826-f026:**
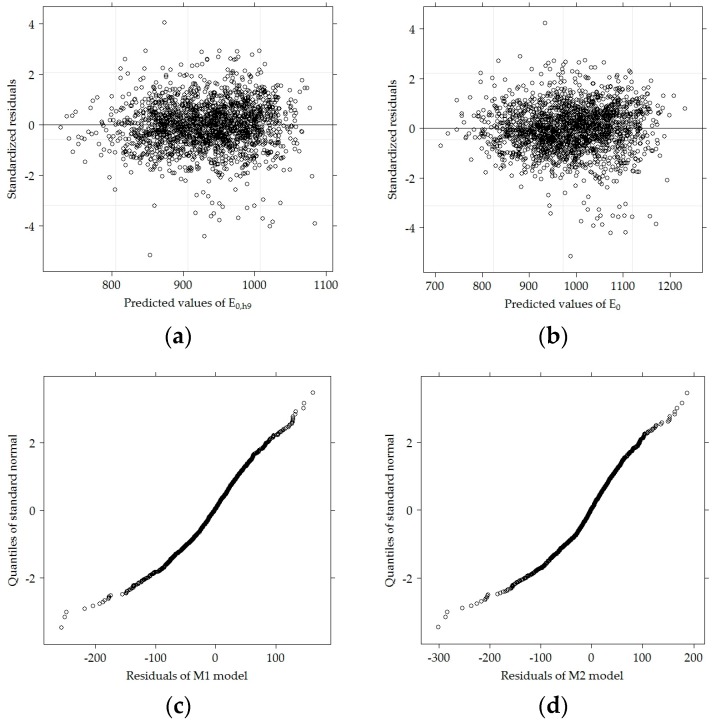
Residual values of models M1 and M2: (**a**) diagnostic of estimated M1 model: predicted values of *E*_0,h9_ versus standardized residuals; (**b**) diagnostic of estimated M2 model: predicted values of *E*_0_ versus standardized residuals; (**c**) diagnostic of estimated M1 model: residuals versus quantiles of standard normal distribution; (**d**) diagnostic of estimated M2 model: residuals versus quantiles of standard normal distribution.

**Table 1 materials-12-03826-t001:** Positions of temperature and moisture content sensors in road pavement structures (PSs).

No of the Road PS	Road PS	Soil of Subgrade (Continuous Measurements)	
	Temperature	Temperature	Moisture Content
No 4	on the top; at a depth of 9 cm	at a depth of 100 cm; 130 cm; 150 cm	
No 12	on the top at a depth of 8 cm; 9 cm; 10 cm	at a depth of 100 cm; 130 cm; 150 cm	
No 18	on the top; at a depth of 8 cm; 9 cm; 10 cm	at a depth of 100 cm; 130 cm; 150 cm	no sensor
No 24–25	on the top; at a depth of 9 cm	at a depth of 100 cm; 130 cm; 150 cm	

**Table 2 materials-12-03826-t002:** Date of deflection measurements in sections.

Measurement No	Date	ESAL_100_	Average Air Temperature during Measurement Time, °C	Average Pavement Surface Temperature during Measurement Time, °C	Average Asphalt Concrete Layer Temperature at a Depth of 9 cm during Measurement Time, °C
1	22.08.2013	484,000	22	20	21
2	10.10.2013	508,000	13	10	12
3	14–15.11.2013	537,000	5	3	5
4	07.05.2014	596,000	12	14	12
5	14.07.2014	621,000	25	31	24
6	16.09.2014	637,000	19	23	18
7	13.11.2014	654,000	7	7	6
8	22–23.04.2015	678,000	9	18	10

**Table 3 materials-12-03826-t003:** Correlation of temperatures and moisture content at different depths of the asphalt concrete layer.

		*E_0_*	*E* _0,h9_	Temperature					Moisture Content			The Thickness of the Asphalt Layer
				At a Depth of 100 cm	At a Depth of 130 cm	At a Depth of 150 cm	On the Top of RPS (Measured Manually)	At a Depth of 9 cm of RPS (Measured Manually)	At a Depth of 100 cm	At a Depth of 130 cm	At a Depth of 150 cm	
*E* _0_		1.00	0.75	−0.38	−0.39	−0.37	−0.53	−0.54	−0.09	−0.12	0.08	0.07
*E* _0,h9_		0.75	1.00	0.25	0.24	0.24	0.03	0.14	0.07	−0.10	0.01	0.12
Temperature	At a depth of 100 cm	−0.38	0.25	1.00	0.99	0.99	0.59	0.89	0.09	−0.17	−0.03	−0.03
	At a depth of 130 cm	−0.39	0.24	0.99	1.00	1.00	0.58	0.89	0.04	−0.21	−0.10	0
	At a depth of 150 cm	−0.37	0.24	0.99	1.00	1.00	0.55	0.87	0.03	−0.23	−0.11	0
	On the top of RPS (measured manually)	−0.53	0.03	0.59	0.58	0.55	1.00	0.84	0.38	0.38	0	0.03
	At a depth of 9 cm of RPS (measured manually)	−0.54	0.14	0.89	0.89	0.87	0.84	1.00	0.23	0.07	−0.12	0.03
Moisture content	At a depth of 100 cm	−0.09	0.07	0.09	0.04	0.03	0.38	0.23	1.00	0.76	0.26	−0.02
	At a depth of 130 cm	−0.12	−0.10	−0.17	−0.21	−0.23	0.38	0.07	0.76	1.00	−0.02	0.03
	At a depth of 150 cm	0.08	0.01	−0.03	−0.10	−0.11	0	−0.12	0.26	−0.02	1.00	−0.16
The thickness of the asphalt layer		0.07	0.12	−0.03	0	0	0.03	0.03	−0.02	0.03	−0.16	1.00

**Table 4 materials-12-03826-t004:** Types of variables in the study.

Dependent Variable	Explanatory Variables		
	Tested PS	Variables Related to the Features of Measured Points in the PS	Time-Varying Properties
*E*_0_ or *E*_0,h9_	Identification number of the tested PS (section)	Identification number of the point in the tested PS. Lane (left wheel path, track and right wheel path). Loading feature (Yes/No). The thickness of the AC layer.	Time (of repeated measurements). Temperature (°C) and moisture content (%) in the soil of the subgrade at a depth of 100 cm, 130 cm and 150 cm. Temperature (°C) on the top of the road PS. Temperature (°C) of the AC layer at a depth of 9 cm.

**Table 5 materials-12-03826-t005:** Estimated parameters of models M1 and M2.

Model Parameters	Estimated Parameters					
	M1 Model			M2 Model		
	Estimate	Standard Error	*p*-Value	Value	Standard Error	*p*-Value
*β*_0_ (intercept)	1002.65	11.19	<0.001	454.96	79.89	<0.001
*β*_1_ (time)	−17.27	1.76	<0.001	−18.72	1.99	<0.001
*β*_2_ (left wheel path)	−21.97	10.04	0.029	−21.31	10.08	0.0356
*β*_3_ (track)	−21.64	10.06	0.0324	−32.13	10.14	0.0017
*β*_4_ (loaded lane)	−88.45	8.21	<0.001	−115.11	8.71	<0.001
*β*_5_ (time × loaded lane)	9.33	1.01	<0.001	10.03	1.08	<0.001
*β*_6_ (time × left wheel path)	14.16	1.24	<0.001	14.99	1.32	<0.001
*β*_7_ (time × track)	15.58	1.24	<0.001	17.09	1.32	<0.001
*β*_8_ (temperature)	-	-	-	−8.41	0.17	<0.001
*β*_9_ (thickness of AC layer)	-	-	-	4.37	0.48	<0.001
*β*_10_ (temperature of the AC layer at a depth of 9 cm × moisture content at a depth of 100 cm)	-	-	-	0.027	0.03	0.391
Standard deviation of random parameters $\sqrt{\sigma^{2}}$ and correlation (elements of matrix D)	σ*_intercept_* among sections = 14.83 σ*_trend_* among sections = 2.57 cor*_intercept_* and trend among sections = −0.516 σ*_intercept_* among points in the section = 48.55			σ*_intercept_* among sections = 18.89 σ*_trend_* among sections = 3.30 cor*_intercept_* and trend among sections = −0.428 σ*_intercept_* among points in the section = 52.19		

Note. *β*_0_−*β*_10_—the value of the parameters estimated without the insignificant interaction of the temperature and the moisture content.

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
