# Peer review of "Influence of Temperature and Moisture Content on Pavement Bearing Capacity with Improved Subgrade"

_materials, 2019, doi:10.3390/ma12233826_

Round 1
Reviewer 1 Report
The paper presents a statistical analysis between the bearing capacity of different asphalt concrete pavements and their thickeness, temperature and moisture content of the layers. The paper is very interesting, appropiate for the Journal, easy to read and clear. The method is scientifically sound and well described. I have just some minor comments:
Page 2, line 51: The authors write that the bearing capacity of pavements "is described by FWD. In my opinion the bearing capacity can be evaluated by FWD, not described.
Page 5 figure 1: It could be better add a legenda. What do VS, AS and PMB mean?
Page 5 table 2, last column: it is not cleae what "Average asphalt concrete layer temperature" means. Is the average for all the sections, in which layer? under the bituminous layers or in the middle?
Author Response
Dear editor and reviewer,
Thank you for your message.
We have carefully read comments of the reviewer to our manuscript and think that these comments are very constructive and have been of tremendous benefit.
In the revised manuscript, we have made some modifications/corrections in accordance with the reviewers’ feedback. Furthermore, we have asked an expert, native English speaker to edit the language. We hope that these changes will help us meet the requirements for publication in your journal.
In the following section, the text shown in red corresponds to new text that has been incorporated into the revised manuscript. All changes (using track change) you can find in attachment.
Thank you for your time and careful work.
Response to Reviewer 1 Comments
Point 1: Page 2, line 51: The authors write that the bearing capacity of pavements "is described by FWD. In my opinion the bearing capacity can be evaluated by FWD, not described.
Response 1: we agree with the comment and have made the change in lines 54-55.
"According to Straube and Jansen [3], the bearing capacity of AC pavements is evaluated by a FWD (the equivalent elastic modulus – E0)."
Point 2: Page 5 figure 1: It could be better add a legenda. What do VS, AS and PMB mean?
Response 2: we agree with the comment and have made the change in lines 191-197.
"AC layers of these sections were constructed from AC11VS, AC16AS, and AC32PS. Herein – AC11WL is asphalt concrete wearing layer, with a mesh size of top sieve of 11 mm and affected by heavy loads; AC16BnL is asphalt concrete binder layer, with a mesh size of top sieve of 16 mm, and affected by heavy loads; AC32BsL is asphalt concrete base layer, with a mesh size of top sieve of 16 mm, and affected by heavy loads. Except one section – No 12, where surfacing performed from the SMA11 PMB WL (stone mastic asphalt wearing layer, with a mesh size of the top sieve of 11 mm, with polymer-modified bitumen, and affected by heavy loads)."
Point 3: Page 5 table 2, last column: it is not cleae what "Average asphalt concrete layer temperature" means. Is the average for all the sections, in which layer? under the bituminous layers or in the middle?
Response 3: we agree with the comment and have made the change in line 221.
"Average asphalt concrete layer temperature at a depth of 9 cm during measurement time, °C".

Reviewer 2 Report
This article evaluates the impact of temperature and moisture content on pavement system from the perspective of experimental design and statistics analysis. The authors have provided as much detailed experimental data as possible and analyzed them reasonably, so I think this article can be published. Please address the following comments:
Please shorten the length of abstract. It’s better to include what kind of experiments conducted for bearing capacity in abstract. Line 60: in statement "The impact of climate change is direct and indirect on flexible pavements” Please change "is" to "has". The authors explaining a lot about weather conditions on pavement which are relevant to the experiments and not necessary to discuss. (like: high temperature or frost) please reduce them. Line 635. In statement “Heavy vehicle loads effect performance of road pavement" please use "affect the performance"
Author Response
Dear editor and reviewer,
Thank you for your message.
We have carefully read comments of the reviewer to our manuscript and think that these comments are very constructive and have been of tremendous benefit.
In the revised manuscript, we have made some modifications/corrections in accordance with the reviewers’ feedback. Furthermore, we have asked an expert, native English speaker to edit the language. We hope that these changes will help us meet the requirements for publication in your journal. All corrections (using track change) you can find in attachment.
In the following section, the text shown in red corresponds to new text that has been incorporated into the revised manuscript. All changes you can find in attachment.
Thank you for your time and careful work.
Response to Reviewer 2 Comments
Point 1: Please shorten the length of abstract.
Response 1: we have made changes in lines 11-36.
"Abstract: Environmental conditions (temperature, moisture, the intensity of the sun) influence variation in asphalt pavement strength during the year. Lithuania is situated in a zone by average warm summers and average cold winters, and the most important climatic factors is variation of the air temperature . This study presents the influence of temperature (of asphalt concrete (AC) and subgrade layers) and moisture content (of subgrade layers) to the pavement bearing capacity. The experimental research was obtained in five pavement sections of the experimental road. This experimental road was constructed in 2007 in Lithuania and is operated more thant 12 years. Paper presents statistical analysis between the bearing capacity and the thickness of the asphalt concrete layers, the temperature and moisture content of different pavement layers, among sections, loaded and unloaded lanes (right and left wheel paths and tracks). The bearing capacity was evaluated by Falling Weight Deflectometer (FWD), temperature and moisture content – by electronic sensors, thickness of AC layers – by Georadar. Analysis of overall E0 and E0,h9 (bearing capacity at a depth of 9 cm from asphalt surface) data declare that seasonal impact on pavement structural strength due to a change of subgrade bearing capacity remains after correction of asphalt stiffness dependent on temperature in layer. However, it was detected that neither E0 nor E0,h9 are related to moisture content at a depths of 100 cm, 130 cm, 150 cm."
Point 2: It’s better to include what kind of experiments conducted for bearing capacity in abstract.
Response 2: we have made changes in lines 23-27.
"Paper presents statistical analysis between the bearing capacity and the thickness of the asphalt concrete layers, the temperature and moisture content of different pavement layers, among sections, loaded and unloaded lanes (right and left wheel paths and tracks). The bearing capacity was evaluated by Falling Weight Deflectometer (FWD), temperature and moisture content – by electronic sensors, thickness of AC layers – by Georadar. "
Point 3: Line 60: in statement "The impact of climate change is direct and indirect on flexible pavements” Please change "is" to "has".
Response 3: we have made changes in line 65. We decided to delete the sentence according to your next comment in Point 4.
Point 4: The authors explaining a lot about weather conditions on pavement which are relevant to the experiments and not necessary to discuss. (like: high temperature or frost) please reduce them.
Response 4: we have made changes in lines 60-72. We decided to delete the information as not necessary to discuss.
Deleted information:
"Climate change affects the properties of the pavement layers, making them more susceptible to accelerated damage under the loading of heavy vehicles, reducing their service life, and increasing maintenance costs. Scientists at NASA Goddard Institute for Space Studies (GISS) state that the average global temperature on Earth has increased by about 0.99 °C since 1880 and at a rate of roughly 0.15–0.20 °C per decade.
The impact of climate change is direct and indirect on flexible pavements. Direct impacts are due to environmental effects, e.g., temperature, precipitation, solar radiation, wind speed, and groundwater level. Indirect impacts refer to changes in the loading of heavy vehicles caused by demographic changes due to climate change [4,5].
As the annual global average surface temperature increases – this presents many challenges for traditional infrastructures, which are usually designed and managed based on historical climate data. In addition, significant changes in the global climate are expected in the coming decades, with an increase in severe weather phenomena such as extreme temperatures and heavy rainfall [6]."
Point 5: Line 635. In statement “Heavy vehicle loads effect performance of road pavement" please use "affect the performance".
Response 5: we have made changes in lines 655-656.
"Heavy vehicle loads affect the performance of road pavement and this effect is more significant (aggressive) in the trajectory of the path of the right wheels. "
